# Women's experiences of anal incontinence following vaginal birth: A qualitative study of missed opportunities in routine care contacts

Joanne Parsons[1]⊚*, Abi Eccles[1]⊚, Debra Bick[2‡], Michael R. B. Keighley[3‡], Anna Clements[3], Julie Cornish[4‡], Sarah Embleton[3], Abigail McNiven[5], Kate Seers[6], Sarah Hillman[1]⊚

1 Unit of Academic Primary Care, Warwick Medical School, University of Warwick, Coventry, United Kingdom, 2 Warwick Clinical Trials Unit, University of Warwick, Coventry, United Kingdom, 3 The MASIC Foundation, Nottingham, United Kingdom, 4 Department Colorectal Surgery, Cardiff and Vale University Health Board, Cardiff, United Kingdom, 5 Nuffield Department of Primary Care Health Sciences, University of Oxford, Oxford, United Kingdom, 6 Warwick Research in Nursing, Warwick Medical School, University of Warwick, Coventry, United Kingdom

⊚ These authors contributed equally to this work.
‡ DB, MRBK and JC also contributed equally to this work.
* jo.parsons@warwick.ac.uk

**Data Availability Statement:** There are no datasets associated with this paper other than transcriptions of interviews with participants which would not be

## Abstract

### Objectives

This study aimed to explore experiences of women with anal incontinence following a childbirth injury, and to identify areas of missed opportunities within care they received.

### Design

This is a qualitative study involving semi-structured interviews.

### Setting

Participants were recruited via five hospitals in the UK, and via social media adverts and communication from charity organisations.

### Participants

Women who have experienced anal incontinence following a childbirth injury, either within 7 years of sustaining the injury, or if they identified new, or worsening symptoms of AI at the time of menopause.

### Main outcome measures

Main outcomes are experiences of women with anal incontinence following childbirth injury, and missed opportunities within the care they received.

shared outside of the research team. Ethical approval for this study does not cover sharing of the data outside of the research team (REC reference number: REC21/EE/0167). HCRW. approvals@wales.nhs.uk.

**Funding:** This project is funded by the National Institute for Health and Care Research (NIHR) under its Research for Patient Benefit (RfPB) Programme (Grant Reference Number NIHR202172). The views expressed are those of the author(s) and not necessarily those of the NIHR or the Department of Health and Social Care. The funders had no role in study design, data collection and analysis, decision to publish, or preparation of the manuscript.

**Competing interests:** I have read the Journal's policy and the authors of this manuscript have the following competing interests: Professor Michael Keighley is Trustee and President of the MASIC Foundation but this has not impacted on conduct of study or publication, CEO of Keighleycolo Ltd involved with medical reports for court in which OASI patients have been seen but no conflict as this simply informed me of the consequences of the injury during one to one face to face consultations. Trustee Friends of Vellore UK also has helped to understand impact of OASI in the Asian community in India. Joint holder of two i4i NIHR grants for a device for the treatment of fistula in fact of value to some OASI patients. This does not alter our adherence to PLOS ONE policies on sharing data and materials.

## Results

The following main themes were identified: opportunities for diagnosis missed, missed opportunities for information sharing and continuity and timeliness of care.

## Conclusions

Anal Incontinence following a childbirth injury has a profound impact on women. Lack of information and awareness both amongst women and healthcare professionals contributes to delays in accurate diagnosis and appropriate treatment.

## Introduction

Anal Incontinence (AI) is the inability to control flatus, faecal urgency (sometimes with urge faecal incontinence) often with post defaecation soiling [1]. Over one in five women will develop AI in the first five years after having a vaginal birth [2]. AI is common [3, 4] and obstetric trauma from pelvic floor, neurogenic and sphincter injury are the most common causes [5]. The timing of symptom onset varies; some women experience AI onset soon after childbirth (which may or may not resolve), whilst many develop worsening or new symptoms during the menopause [6].

Obstetric anal sphincter injury (OASI) (damage to the anal sphincter that occurs during vaginal births) detected at the time of birth (a cause of AI) is reported to occur in approximately 3.5% of vaginal births in the UK [7] and in 6% of nulliparous women (women who have never had a live birth before) [8, 9]. An instrumental birth, especially involving the use of forceps [10] raises the risk of OASI injury further still [9]. Maternal age is a significant risk factor [11], as is higher weight of the baby [12]. Women of Indian and Pakistani origin are significantly more likely to sustain an OASI during childbirth [13–15], the reasons behind which are likely to be complex and multifactorial [16]. Initiatives to reduce the risk of OASI have been introduced in the UK such as the OASI care bundle (Royal College of Obstetricians & Gynaecologists and Royal College of Midwives), which includes a set of interventions to improve outcomes, such as informing the woman about risk of OASI and the use of manual perineal protection at the time of birth whenever possible [17, 18].

Incidence of OASI has increased threefold in the last decade [8, 9] especially following a first vaginal birth [19]. Whilst some of this may be attributed to better detection, risk factors such as maternal age and baby weight have also increased all of which may have likely attributed to the rise in incidence.

AI has a significant negative impact on quality of life. There are undesirable and often life-changing effects on psychosocial and emotional wellbeing as women may struggle to achieve their basic activities of daily living [20] and are more likely to experience post-traumatic stress [21]. Whilst AI shares characteristics with other stigmatising health conditions, faecal matter and odours are a particularly taboo form of 'matter out of place' [22] and healthcare professionals often focus more on urinary incontinence than AI [7]. The focus on urinary incontinence over AI is likely because the condition is less stigmatising and more commonly discussed.

Due to the stigmatising nature of AI, it is often not reported by those who suffer it, with less than 25% of women discussing their problems with GPs unless prompted [23]. Healthcare professionals (HCPs) may feel uncomfortable asking questions about AI in postnatal encounters,

or lack experience of asking, detecting and managing the condition. This represents a missed opportunity to identify and refer women to appropriate care pathways [24–27].

Findings of the recent Ockenden review from 2022 (an independent review of maternity service provision at one NHS Trust in England, commissioned by NHS improvement) highlighted a lack of clinical training, governance and leadership when reviewing the antenatal, intrapartum and immediate postnatal care of women who gave birth at the Trust [28]. Many of the themes around a lack of clinical training, governance and leadership exist in the longer postnatal period, beyond that captured by the independent reviews. In 2022, the Women's Health Strategy for England reported findings from a survey of over 97,000 individuals and highlighted a lack of appropriate and timely care in general for women [29].

This study aimed to explore the experiences of women experiencing anal incontinence since sustaining a perineal birth injury, and to identify areas of missed opportunities within the care they received. This aim was achieved via interviews with 41 women.

## Methods

This research is reported using the Standards for Reporting Qualitative Research framework [30]. Data was collected using semi-structured interviews with women who have experienced anal incontinence following a childbirth injury.

### Ethics

Ethical approval was obtained from the East of England—Essex Research Ethics Committee (REC21/EE/0167).

Informed consent was obtained prior to any interviews taking place. Consent was either written or verbal depending on participant preference. Verbal consent was audio recorded and saved separately to interview files.

### Sampling and recruitment

Participants were recruited to the study between October 2021 and May 2022. Participants were eligible to participate if they met the following criteria:

1. Women aged 18 and over

2. AI as a result of obstetric injury (symptoms apparent within 7 years of vaginal birth) or AI at time of menopause for whom vaginal birth is the likely contributing factor

    Participants were not eligible to participate if they:

1. Experienced severe mental health or learning difficulties meaning that they cannot consent to take part

2. Had no history of vaginal birth

    Participants were recruited via hospitals or related organisations networks, such as The MASIC Foundation.

    Participants made contact directly with the research team, following adverts for participants being posted on the social media accounts of either the medical school where the research was based or groups/charities aligned with the work (Facebook and Twitter), and via social media and known networks available to The MASIC Foundation www.masic.org.uk.

    Participants were also recruited via five hospitals across the UK (Cheshire, East Midlands, North West England, Wales and West Midlands). Women were informed about the study and willing participants completed a form consenting to contact details being passed to the

research team who subsequently contacted the women. Aiming for a maximum variation sample to capture a range of views and experiences, women were recruited with varied backgrounds according to ethnicity, deprivation (measured by postcode and calculated using the Index of Multiple Deprivation [31] and the Welsh Index of Multiple Deprivation [32]) and whether they were within seven years of the birth injury or around the time of menopause. During fieldwork, a large proportion of the initial sample were white British, and subsequent recruitment strategies were modified to purposively sample non white British participants. Strategies included translation of adverts into different languages, recruitment through the social media accounts of groups/charities aligned with improving the health of ethnic minority women and targeted recruitment in areas of higher rates of non-white British populations.

## Patient and public involvement

Patient and Public Involvement (PPI) has been included throughout the study, from being involved in discussion at the initial design of the study, through the funding application process by providing input and feedback into drafts of the application and throughout the study itself by attending meetings to discuss progress, help resolve any difficulties and provide a patient perspective at each stage. PPI input has been obtained on the study design, study materials (particularly interview schedules), recruitment, analysis and dissemination. PPI input has taken the form of small group and one-to-one discussions with women with lived experience of AI following a childbirth injury. PPI input ensured what we were asking women was appropriate, and they ensured the interviews were covering the main issues, and helped the research team to consider things from the perspective of women with lived experience of AI.

## Data collection

Semi-structured interviews were conducted via either Microsoft Teams or telephone (depending on the preference of the participant). Experiences of the original injury, contact with healthcare professionals, the impact of the injury on everyday life, relationships with family and friends, work and finances, and information needs were all discussed.

Topic guides were designed with input from existing literature, healthcare professionals and public representatives (women with injuries) and were further developed iteratively as interviews progressed. Interviews were conducted by two experienced qualitative researchers and were recorded and transcribed verbatim using a combination of Teams automated transcription and a professional transcription service. All transcripts were checked against the audio for accuracy.

Only the immediate research team (JP, AE and SH) had access to participant personal information upon recruitment. Other authors had access to anonymised data only.

## Data analysis

Interviews were analysed using Thematic Analysis, as described by Braun and Clarke in 2022 [33], which provides a flexible approach to qualitative data analysis. The coding frame was developed as the research team became familiar with the data and was revised iteratively as analysis progressed.

Braun and Clarke defined Thematic Analysis with six steps, and these were applied to the data:

1. Familiarising yourself with the dataset;

2. Coding;

3. Generating initial themes;

4. Developing and reviewing themes;

5. Refining, defining and naming themes; and

6. Writing up

In line with the six steps outlined above, the members of the research team most familiar with the data (JP, AE, SH) read a sub-section of transcripts (step 1) and discussed initial codes that were apparent in the data. This led to an early draft of the coding framework (a list of codes) which these initial transcripts were coded with (step 2). Coding was discussed with the wider research team, and changes were made in line with this discussion. Further transcripts were then coded using the list of codes, and consistency of coding was checked to confirm researchers were coding in a similar way. The remainder of the transcripts were then coded, and a list of themes were generated and developed (steps 3 and 4) and shared with the wider team for approval, resulting in a final set of themes that accurately represented the data collected (step 5). These themes were then included in the write up of the study.

## Results

A total of 41 participants were recruited. Of these, 15 were recruited via hospital recruitment sites, and 26 were recruited via social media advertising. Table 1 shows characteristics of participants.

Three main themes have been identified, each with subsequent sub-themes. These are opportunities for diagnosis missed (sub-themes: normalisation; feeling dismissed; prioritisation of baby's needs; lack of examination; and incorrect diagnosis), missed opportunities for information sharing (sub-themes: feeling ill informed about injury; lack of debrief or unanswered questions after birth; lack of awareness of AI; focus on bladder rather than bowel symptoms; linking to information opportunities antenatally and ack of information on the link between injury and menopause), and continuity and timeliness of care (sub-themes: inadequacy of the six-week check; waiting lists and chasing services; stuck between specialisms and mental and emotional needs ignored), Links between the themes are evident and demonstrate the complexity of women's experiences. Additional quotes for some themes can be found in Supplemental material (S1 Table).

### 1. Opportunities for diagnosis missed

Women reported a number of factors which appear to contribute to an absence of an accurate and timely diagnosis including Normalisation; Feeling dismissed; Prioritisation of baby's needs; Lack of examination; and Incorrect diagnosis.

**Normalisation.** Giving birth is often considered a physically traumatic event in itself, and a period of healing is expected for all women. This appears to have led to confusion, for women and healthcare professionals, as to whether symptoms were a 'normal' result of labour and birth which would improve with time, or something that required healthcare input. Participants reported how they normalised what they were experiencing and attributed their symptoms to expected outcomes of childbirth.

*Yeah so, what happened with me was that during childbirth I had, I think it was the third degree tear I had during childbirth. And in the days and weeks that followed, I was still in a lot of pain but I just assumed I've just had a baby, that's why, and I thought no other mums are complaining about it, so I should just be getting on with it as well (P32, post-natal, 35–44).*

**Table 1. Demographics of participants.**

| Characteristic | Within 7 years of childbirth (n = 25) | Menopause (n = 16) |
|---|---|---|
| **Mean age (range)** | | |
| | 36.04 (20–47) | 54.75 (41–75) |
| **Recruitment source n (%)** | | |
| Social media | 14 (56) | 12 (75) |
| Hospital recruitment site | 11 (44) | 4 (25) |
| **Ethnicity n (%)** | | |
| White British | 17 (62) | 14 (87.5) |
| White Polish | 2 (8) | |
| White other | 2 (8) | |
| British Asian | 3 (12) | |
| British Pakistani | | 1 (6.25) |
| British Indian | | 1 (6.25) |
| Black Caribbean | 1 (4) | |
| **Marital Status n (%)** | | |
| Single | 1 (4) | 1 (6.25) |
| Married | 17 (68) | 14 (87.5) |
| Divorced | | 1 (6.25) |
| Co-habiting | 7 (28) | |
| **Parity** | | |
| **Deprivation score n (%)*** | | |
| 1st quintile | 6 (24) | 2 (12.5) |
| 2nd quintile | 5 (20) | 7 (43.75) |
| 3rd quintile | 3 (12) | 2 (12.5) |
| 4th quintile | 7 (28) | 3 (18.75) |
| 5th quintile | 4 (16) | 2 (12.5) |

* Deprivation was calculated based on postcode, and converted to a deprivation score using the Index of Multiple Deprivation and the Wales Index of Multiple Deprivation. A score of 1 is least deprived and a score of 5 is most deprived.

Many women talked about how healthcare professionals often gave the impression that their symptoms were a normal consequence of labour.

*I felt that GP had no appreciation of what's happened, no disrespect to GP. Uh, but I think they had no appreciation and when I had my six week check, it was very general questions. They're very general, and even when I've addressed things like that, they didn't do anything with it in a way. They just said it's pretty normal, it's pretty standard. You know it's only six weeks, but then I wasn't offered anything beyond that. Sometimes I feel like, within the six weeks, everyone can tell you that whatever happens to your body, it's normal, because everything has been thrown out. And that's fine, but when is the moment to start addressing. . . well, actually post this time I still feel that something is wrong and is it normal or is it not normal? And I feel with an injury like that, I think that should be addressed definitely (P33, postnatal, 35–44).*

This normalisation meant many women were unsure if everyone experienced such symptoms after having a baby. This led many to think there was no support available and sometimes they were reluctant to seek subsequent help from healthcare professionals.

**Feeling dismissed.** Women's reluctance to ask for help with symptoms of AI were further compounded by reported feelings that healthcare professionals were often dismissive. They described how symptoms were often not taken seriously, and this often led to participants not continuing to seek help for their AI symptoms. This feeling of not being taken seriously or dismissed is evident across healthcare professional types, with some women reported still feeling dismissed once referrals had been made to secondary or tertiary care, suggesting women are feel their symptoms are dismissed throughout their care journey.

*The colorectal surgeon was very dismissive, didn't really care you know, he sort of had the attitude here this has happened and you just have to get on with your life, and that was it (P13, post-natal, 35–44).*

**Prioritisation of baby's needs.** Participants often felt that their babies' needs were prioritised over their own. Women often felt that it was not just healthcare professionals who prioritised the babies' needs, but they themselves did too. Participants described healthcare professionals being more focused on their baby, particularly at the six-week GP check, resulting in their own needs being overlooked or women not mentioning them.

*But professionals that are involved should be picking up on this as well. The, the people that are going out there, not just there to see if that baby is gaining weight. . .. It shouldn't be the only thing of any importance for me. . .. Is your baby gaining weight? Yes. Goodbye. Not about how are you. Are you OK? You know (P10, menopause, 45–54).*

*And you think like really obviously just had a baby you're trying to? You know, yeah, you're so tired trying to breast feed all this, that and the other and then you've got that on the back of it. You know. So like you're trying to put your focus on the baby aren't you? But then, if you know realistically if it was 'cause of something else and that make you feel like the focus was more, on yourself. You know it's a lot. It's kind of a lot to take in at that time, isn't it? And you just come last in that scenario, you're kind of grateful aren't you that your baby is well etcetera. (P11, post-natal, 25–34).*

**Lack of examination.** Participants discussed how they were often not examined after sustaining an injury during childbirth (sometimes immediately after birth, and sometimes later in the postnatal period as well), leaving the injury unidentified and causes of anal incontinence unknown, thus missing the opportunity for a timely and accurate diagnosis.

*When I went to my GP for my post- check up thing, didn't even examine me. I said I'm having a really bad time, I don't think things are right then just said 'it's like that for lots of ladies. Just get on with it.' And that was it really (P10, menopause, 45–54).*

*So when I had, when the injury occurred. First of all, I would have liked it if she did notice that I had internal damage because I needed to be stitched by a colorectal surgeon and I would not have any issues at all. So that was a missed diagnosed tear (P4, menopause, 45–54).*

**Incorrect diagnosis.** Women often reported a delay in obtaining an accurate diagnosis, leaving them without support or confirmation of what was causing the symptoms of AI. Some participants reported receiving an incorrect diagnosis, again delaying the receipt of appropriate support or treatment. Many women had been told their symptoms were a result of Irritable Bowel Syndrome (IBS) and were given treatment or advice for this incorrectly.

*I just went to the GP 'cause I was literally on the toilet all the time. . . . multiple times a day, and I asked her and she just said it's just IBS, "You've got IBS". . . . which I obviously went away and thought, "No, I'm not happy with that." So I sort of keep going back and asking, and she just kept telling me it was IBS and, and this went on probably for about three years altogether (P27, menopause, 55–64).*

*It was a midwife from that ward coming round doing the checks and I mentioned it to her and she said I will treat it for constipation 'cause that can cause like an overflow sometimes. So they did that. They treated it as a constipation for about six weeks, but nothing ever came of it 'cause I, I personally would say I wasn't constipated, they just kind of assumed that (P20, post-natal, 18–24).*

For many participants, normalisation of symptoms and feelings of being dismissed by healthcare professions results in them not knowing where to go for support and treatment. Therefore, they are less likely to receive appropriate examinations and diagnosis. Confusion with other conditions further complicates the situation.

### 2. Missed opportunities for information sharing

Women reported various situations and timepoints where information on symptoms or injuries was incomplete or entirely absent. The timing of the information was also important in helping women to feel adequately informed and able to fully understand the injury and the symptoms they experience. Participants reports a number of ways opportunities for information sharing was missed: Feeling ill informed about injury; Lack of debrief or unanswered questions after birth; Lack of awareness of AI; Focus on bladder rather than bowel symptoms; Linking to information opportunities antenatally and Lack of information on the link between injury and menopause.

**Feeling ill informed about injury.** Participants often reported feeling ill informed, lacking information about what had happened during childbirth and what the consequences of injuries were.

*Well I didn't know I'd had an injury. . .. I didn't really. I mean I knew, I sort of knew, because I knew this isn't right but I didn't know at that point exactly what had happened. (P10, menopause, 45–54).*

*But that's about as much as I know, like she never really said, I mean, I can assume, I know sort of where I torn, but that's all I knew (P14, post-natal, 25–34).*

**Lack of debrief or unanswered questions after birth.** Many participants reflected on how they would have valued a labour debrief to understand what happened and the injury they sustained. Many participants reported feeling like they were not given the information needed about what injury had occurred and how tears had been repaired, or described feeling that they had been left with unanswered questions about some of the decisions made during labour. For example, some questioned if they should have had a c-section; if they were advised to push for too long; if they should not have had their waters broken; or if a consultant should have come sooner.

*'Cause obviously I had stitches. And then, you know, they would check them. But there was no, I never had, "This is what we did," . . . you know, I'm presuming it's in my notes somewhere. But I, you know, I've never been told, "It was this," you know, "This is how many*

*stitches you had." You know, there was none of that. We just didn't, it just wasn't mentioned. It was like, "We've stitched you up. Off you go" (P12, menopause, 45–54).*

*I wonder whether some of my issues could have been avoided if I had had a C-section, be that elective or emergency one, but earlier. Because a lot of the issues are caused by the forceps and, the long labour (P30, post-natal, 35–44).*

Participants reports show clear evidence that they felt ill informed and would find a debrief after labour useful.

**Lack of awareness of AI.**    There was a perception from participants that there was a lack of awareness amongst healthcare professionals, and society more widely, about AI as a result of a childbirth injury. This was felt to be a barrier to receiving appropriate care and support, and also to feeling that they could talk about their symptoms and what they were going through.

*I don't think, even in the hospital it was like it was like it'd never happened before. It was like it was as if they were dealing with. I don't know like. Strange. I mean one comment that the physio did say to me when I did end up having the appointment was that if it's not talked about enough so people don't know it happens. It's more common than you think, but also. The women go home, they might not tell you about it (P11, post-natal, 25–34).*

*I wish it was more talked about. I wish society understood it better you know, and knew that this was a thing you know, so, 'cause I think other women would . . . you know, I'm pretty sure the other women are there but like me, they probably don't even tell . . . you know, I don't even tell my sisters never mind you know, the outside world (P16, menopause, 65+).*

**Focus on bladder rather than bowel symptoms.**    Closely linked to reported lack of awareness of AI, participants also reported an emphasis on knowledge and care on bladder issues. This may be because many women and healthcare professionals are more comfortable discussing bladder problems than bowel issues, and also because of the higher prevalence of postnatal bladder issues. Some participants experienced both urinary and anal incontinence, but the focus and the support offered was more focused on urinary incontinence.

*I was starting to sort of be aware that sometimes my my bowels were not functioning as I sort of thought they were, but it was something because the focus was always on my bladder, that it got sort of left, but it wasn't until like 2016. I, I was in sort of menopause, premenopausal sort of age, I was 48 then and I was aware that things were not as they should be (P7, menopause, 45–54).*

*There, there just is no information, or you know when. When I went through it, you know, even looking online I could not find any information. It was all about pelvic floor and weeing and jumping on trampolines. And but there was nothing at all. So I just felt like I was completely like I was the only person in the world that had ever happened to like, there was absolutely nothing (P8, menopause, 45–54).*

**Linking to information opportunities antenatally.**    Some participants felt that all pregnant women should be provided with information about AI and the risks before giving birth, as part of routine antenatal information offered to women. Increased awareness and knowledge antenatally might help women understand when symptoms are not just expected results of labour and birth, but where more intervention and support from healthcare professionals is

needed, and when to ask for help. It would also help women planning their births, and making important decisions around options and risks involved in delivery.

> *Would've liked . . . it, it would've been nice if I'd known when I was going in to have, have my baby what the, what the possible consequences of a bad tear would've been so that I maybe could've sort of realised that it wasn't a normal symptom to have ongoing incontinence issues, does that make sense? (P13, post-natal, 35–44)*

> *Or I don't know like the midwife could give you stuff during those pre birth checks. You know, you know they give you, they give you information on breastfeeding, but there's very little information about, you know (P30, post-natal, 35–44).*

## Lack of information on the link between injury and menopause

Many participants who had experienced either the onset or a worsening of symptoms during the time of their menopause reported that they were unaware that this could happen or that there was a link. A need for information that AI symptoms can be triggered by the onset of menopause would have been helpful in understanding what was happening, and potentially seeking help earlier.

> *They've been bad for the majority of the time, but I would say that they are worsening with menopause and and I didn't realise that that can happen. Women that had injuries but never had any issues with their injury. They can now get issues in the menopause, so I didn't realise that, so that's why mine are a bit more worse (P4, menopause, 45–54).*

> *So I had to go to see a gynaecologist. And I think I might mentioned then about the incontinence thing, and it was, that was probably the first time I'd heard someone say, 'there's lots of women of your age going through menopause that are presenting with these problems' and that was out of the blue (P7, menopause, 45–54).*

### 3. Continuity and timeliness of care

Women report a number of ways that content or timing of contact with healthcare professionals impacts on the care journey they encounter when dealing with AI. This includes; Inadequacy of the six-week check; Waiting lists and chasing services; Stuck between specialisms and Mental and emotional needs ignored.

**Inadequacy of the six-week check.** The six-week check was often reported by women as not being appropriate for their needs. It is likely that for many women six-weeks post birth is too early to be ready to disclose symptoms; particularly those who felt what they were experiencing was a normal part of labour recovery. However, some women felt that it would have been a good opportunity to be examined and asked about any changes or difficulties with controlling bowel movements or flatus. This may have allowed problems to be detected and women could be signposted or referred on for further help.

Some women said they were not offered a six-week check, and those that did often reported that it was focused only on checking on the needs of the baby, and that their own needs were neglected or not able to be addressed in the short time available. Participants who were not offered a maternal six-week check often felt they would have benefited from one.

> *Well, like like I think it's, I don't know just from one thing I think is absolute madness is that, GPs don't do a six week check on mums that I think that's absolute madness. That seems to me like something that could easily be changed and would do a lot of good because there's so*

*many things that could be worrying you or, causing you problems at that stage (P30, post-natal, 35–44).*

*I would investigate post postnatally or maybe even to put that into the six week check, whenever women have experienced tear and what a degree of tear they've experienced. And maybe focus a little bit more about that, because they are focusing on their like mental questionnaire and there's that but there's so many things to go through, including child. (P33, post-natal, 35–44).*

Women's accounts suggests that six-weeks checks are viewed as important, but not necessarily adequate or at the most appropriate time. Postnatal checks for women, beyond the six-week check would be useful in ensuring women with AI following a child-birth injury are identified and appropriate action is taken.

**Waiting lists and chasing services.** Some women reported having to wait for a long time for appointments for consultations, treatment or surgery. Women also reported having to actively chase up referrals or appointments after feeling that things were taking a long time or not progressing. This left women with long periods of time where they were left to manage symptoms alone, or struggle on with no support. Waiting times were amplified during Covid-19, but these waits existed before the start of the pandemic.

*It took . . . from seeing the gynaecologist and, and getting the scan just before I seen the colorectal surgeon there was a year gap on the waiting list just to get the scan, yeah (P13, post-natal, 35–44).*

*I had to wait over a year for* [Percutaneous Tibial Nerve Stimulation] *PTNS therapy, you know? And in that time, God, like, it was just awful, that waiting period that you have to put things in, in place because obviously it is NHS, you have got the waiting times and, especially now with Covid and stuff, and, you know, the more, I think the more people know about these pessaries as well, they, they're, supposedly one of the women on the group that I'm on, on Facebook, said that there is a shortage of pessaries at the moment (P15, post-natal, 25–34).*

**Stuck between specialisms.** Some women reported difficulty in navigating specialists and referrals. For some this was reflected by the experience of being referred to the wrong person or the wrong speciality, prolonging the time and frustration in reaching appropriate treatment or information. This may be closely linked to the lack of knowledge and awareness of AI amongst healthcare professionals. This problem is further complicated by the structure of many systems and departments. Women reported being referred to different consultants for different parts of their treatment, seeing two different specialists for the same problem, such as a colorectal surgeon and gynaecologist. This increased the time taken to fully address the issues, and also made the process more complicated for women trying to navigate it.

*They're getting posted from different places like you have to see one person for your if it's a vaginal prolapse, you have to see another person if it's a, if it's a rectal prolapse, you have to go see somebody else with something else. It's just. . . You know and then the time that it takes (P19, menopause, 55–64).*

*I have now two separate consultants, one is a colorectal consultant who specialising in a birth trauma. And then a second one is a gynaecologist that's specialising in birth trauma. And, obviously, colorectal is from end and then everything at the front until the middle is the first one. And I did not know how to do a, kind of, narrow specialisations. And one doesn't really look at what the other issue is, it's a bit, you know? (P1, post-natal, 35–44)*

**Mental and emotional needs ignored.**   Many participants reported feeling like their mental and emotional needs were overlooked. Many women reported struggling to cope with symptoms of AI and the impact that that had emotionally, but felt that there was a lack of information on sources of support to support them through the difficult experiences they are facing.

*Probably from like a mental health perspective as well. I don't know if it's one to go into depth as such, but just having those again just links or resources that or people that women can talk to about their experiences (P25, post-natal, 35–44).*

*But it has got worse, it just, yeah, the longer the time's going on, a, the worse my prolapse is and, b, the worse my, my mental health is as well (P27, menopause, 55–64).*

## Discussion

### Summary of main findings

Women reported missed opportunities in respect to the process of obtaining a diagnosis (including a lack of examination, delayed or missed diagnosis and normalisation or dismissal of symptoms), and missed opportunities within the provision of appropriate and timely information.

It is evident that AI following a childbirth injury is hugely traumatic for some women, and some delay to diagnosis might be expected, however the delays experienced by some women is unacceptable. A recent report on pelvic floor services reported that it can take patients up to ten years to get an accurate diagnosis and appropriate treatment [34]. Delays of this magnitude would not be acceptable in other health conditions or injuries, particularly following a traumatic delivery with the added responsibility of caring for a new-born baby. Appropriate and timely aftercare and support (including the provision of mental health support) is needed, and should be available following a childbirth injury.

The bias towards a focus on bladder issues and urinary incontinence over anal incontinence is partly due to the structure of services. Urinary and anal incontinence are treated by completely separate departments (urogynaecology and colorectal departments), with most colorectal departments not having routine access to physiotherapy. This means services remain separate and do not interact with each other. The disjointed process of multiple referrals across different specialities could be overcome by the provision of recognised pelvic floor clinics [35].

### Strengths and weaknesses

The strengths of this study include that this is the first known study to explore the experiences of women who have experienced AI following a childbirth injury in depth, and the missed opportunities in the care women receive. A further strength is the valuable knowledge about experiences of women with AI after childbirth this study creates. It outlines their information and care needs, and highlights deficiencies in care following a childbirth injury.

Strengths of this research include the large sample that was recruited for this study, with 41 participants. We were successful in recruiting a diverse sample including women from different ethnicities and different socioeconomic backgrounds and we were able to obtain views on different stages of women's healthcare journeys. We spoke to women both within 7 years of childbirth, and during the onset of menopause. This allowed us to identify different timepoints where information and care is needed. The themes are present across women's accounts, and across different geographical regions and are important to consider.

Limitations of this study includes the fact that it reports findings from interviews with women about their experiences of AI following a childbirth injury. It is acknowledged that these findings are based on women's accounts alone, and therefore presents their view of what has happened. This study does not provide insight into healthcare professionals' perspectives in these situations.

This study was conducted during the Covid-19 pandemic. Whilst for some women in this study, Covid-19 reduced the need to leave home and increased the acceptability of working from home (and therefore being nearer to the toilet) for others it exacerbated struggles that women reported, leading to less availability of face-to-face appointments, and longer waiting times for some appointments and surgery. However many women experienced these issues a long time before Covid-19 and so whilst exacerbating some issues, it does not account for the struggles and delays women experienced.

## Comparison with previous literature

There have been other qualitative studies with women who have suffered from severe perineal trauma resulting in anal incontinence which have highlighted the difficulty that these women have in talking about their symptoms, their feeling of isolation, the impact that this has on body image and the psychosexual morbidity [23, 36, 37]. In this study there was a saturation of themes regarding missed opportunities with respect to awareness of potential injury, failure of detection at birth, early postnatal recognition as well as delayed diagnosis and treatment. We believe that this failure to grasp opportunities to reduce the morbidity of OASI has not been previously reported from a representative cohort of those with injury.

This study is in line with findings from the Women's Health Strategy for England which included findings from a survey of over 97,000 individual, 84% of whom said there were instances when they had not been listened to by HCPs, and a significant number of women whose gynaecological symptoms were "normalised" rather than acknowledged and dealt with appropriately [29].

## Implications for practice and research

The study's findings suggest AI awareness due to childbirth injury is suboptimal amongst women, and potentially lacking for healthcare professionals too. Information about the potential for AI is required at all stages of the maternal care pathway (including antenatally, during labour, post injury and at the onset of menopause). This is still not happening despite the Montgomery ruling [38] which enforces patient autonomy via open and clear communication between patients and healthcare professionals, requiring healthcare professionals to advise patients fully of risks and alternatives to care [38].

Women within this research commonly reported that their AI had been misdiagnosed as Irritable Bowel Syndrome (IBS). Some of the symptoms appear similar, but the causes and the treatment differ. More work is needed to highlight the potential misdiagnosis and how this risks delayed access to effective treatment and support. It is not clear whether the reported misdiagnoses are a consequence of the lack of HCP training, but strategies to raise awareness of AI following childbirth injury are required both for the general public and HCPs [39].

Further work should look at experiences of women that do not get as far as accessing care for support of AI after childbirth injury. These women are a disadvantaged group likely to be from socio-economically-deprived backgrounds and will require additional support to access care pathways.

Once accessed, care pathways for women with AI after childbirth require optimisation, to minimise variation across regions and provide women with appropriate, timely and individualised care.

Joined up and integrated gynaecology and colorectal services would reduce delays in the diagnosis and treatment of anal incontinence, and provide more holistic care without the need for repeated referrals to different specialisms. Women's health hubs in England offer the opportunity for care to be more integrated, to improve experiences and outcomes for local populations, address inequalities and reduce costs. However interim reports suggest that most areas do not yet have a women's health hub and that current models are diverse and complex [40]. Further review of these models should consider if AI services are being included in current hub models.

## Conclusion

AI after childbirth injury has a profound impact on women. Lack of information and awareness of AI both amongst women and healthcare professionals in addition to complex and unclear pathways contributes to unacceptable delays in accurate diagnosis and appropriate treatment.

Further work to create equitable care pathways for AI after childbirth injury needs to be established urgently with focus on socioeconomic and ethnicity factors.

Supplemental material can be found in S1 Table of additional quotes

## Supporting information

**S1 Table. Additional quotes.**
(DOCX)

## Acknowledgments

The authors would like to thank Jen Hall from the MASIC Foundation for her support throughout the study.

## Author Contributions

**Conceptualization:** Joanne Parsons, Abi Eccles, Debra Bick, Michael R. B. Keighley, Anna Clements, Julie Cornish, Sarah Embleton, Abigail McNiven, Kate Seers, Sarah Hillman.

**Formal analysis:** Joanne Parsons, Abi Eccles, Debra Bick, Kate Seers, Sarah Hillman.

**Funding acquisition:** Joanne Parsons, Abi Eccles, Debra Bick, Michael R. B. Keighley, Julie Cornish, Sarah Embleton, Abigail McNiven, Kate Seers, Sarah Hillman.

**Investigation:** Joanne Parsons, Abi Eccles, Debra Bick.

**Methodology:** Joanne Parsons, Abi Eccles, Debra Bick, Kate Seers, Sarah Hillman.

**Project administration:** Anna Clements.

**Resources:** Michael R. B. Keighley, Anna Clements, Julie Cornish, Sarah Embleton, Abigail McNiven, Sarah Hillman.

**Supervision:** Debra Bick, Michael R. B. Keighley, Kate Seers, Sarah Hillman.

**Validation:** Joanne Parsons, Abi Eccles, Michael R. B. Keighley, Sarah Hillman.

**Visualization:** Joanne Parsons, Abi Eccles, Abigail McNiven, Sarah Hillman.

**Writing – original draft:** Joanne Parsons, Sarah Hillman.

**Writing – review & editing:** Joanne Parsons, Abi Eccles, Debra Bick, Michael R. B. Keighley, Anna Clements, Julie Cornish, Sarah Embleton, Abigail McNiven, Kate Seers, Sarah Hillman.

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
