## [Decision Letter · Decision Letter 0]

24 Apr 2023

PONE-D-23-06127Women's experiences of anal incontinence following vaginal birth: a qualitative study of missed opportunities in routine care contactsPLOS ONE

Dear Dr. Parsons,

Thank you for submitting your manuscript to PLOS ONE. After careful consideration, we feel that it has merit but does not fully meet PLOS ONE’s publication criteria as it currently stands. Therefore, we invite you to submit a revised version of the manuscript that addresses the points raised during the review process.

We look forward to receiving your revised manuscript.

Kind regards,

Antonio Simone Laganà, M.D., Ph.D.

Academic Editor

PLOS ONE

2. In the ethics statement in the Methods, you have specified that verbal consent was obtained. Please provide additional details regarding how this consent was documented and witnessed, and state whether this was approved by the IRB

“I have read the Journal's policy and the authors of this manuscript have the following competing interests:

Professor Michael Keighley is Trustee and President of the MASIC Foundation but this has not impacted on conduct of study or publication, CEO of Keighleycolo Ltd involved with medical reports for court in which OASI patients have been seen but no conflict as this simply informed me of the consequences of the injury during one to one face to face consultations. Trustee Friends of Vellore UK also has helped to understand impact of OASI in the Asian community in India. Joint holder of two i4i NIHR grants for a device for the treatment of fistula in fact of value to some OASI patients.”

Additional Editor Comments:

The reviewers have expressed positive comments regarding your article, raising only few concerns. Considering this point, I invite authors to perform the required minor revisions.

Reviewers' comments:

Reviewer's Responses to Questions

**Comments to the Author**

1. Is the manuscript technically sound, and do the data support the conclusions?

Reviewer #1: Yes

Reviewer #2: Yes

Reviewer #3: Yes

2. Has the statistical analysis been performed appropriately and rigorously? 

Reviewer #1: Yes

Reviewer #2: I Don't Know

Reviewer #3: No

3. Have the authors made all data underlying the findings in their manuscript fully available?

Reviewer #1: Yes

Reviewer #2: Yes

Reviewer #3: No

4. Is the manuscript presented in an intelligible fashion and written in standard English?

Reviewer #1: Yes

Reviewer #2: Yes

Reviewer #3: Yes

5. Review Comments to the Author

Reviewer #1: Overall a well-structured, appropriately designed and original piece of research. Importantly, the value of this research in terms of the existing issue are set out clearly and potential avenues for improving experiences of women after childbirth injury are suggested without breaching the capabilities of qualitative research. An overview of the themes and subgroups in each would add clarity before the main text of the results.

Reviewer #2: PLOS ONE

Women's experiences of anal incontinence following vaginal birth: a qualitative study of missed opportunities in routine care contacts

Reviewer comments

Title P2

appropriate to the paper

Abstract P2

Clear, focused, check for unnecessary capitalisation in the results section.

Intro P3

Line 54 ‘nulliparous women’, could the authors add a brief explanation of this term please,

Line 55 ‘OASI injury’ could the authors add a brief explanation of this term please

Line 56 ‘Women of Indian and Pakistani origin are significantly more likely to sustain an OASI during childbirth (13,14, 58 15).’ Could you say briefly why this is please

Line 61, could you add a sentence adding the implications please.

Lin 67, could you add a sentence to say why this is the case please, it’s a fascinating study, and not everyone will be a medical professional, or familiar with the field.

Methods P4

P4 Line 73 date needed please

P5 line 99 ‘Participants were also recruited via five hospitals across the UK’ could you say which countries, regions please

P5 line . ‘Aiming for a maximum variation sample, women were recruited with varied backgrounds according to ethnicity, deprivation’ could you say why you wanted a variation sample, and what type of deprivation/how you were assessing deprivation please,

P5

Line 110 the PPI section is interesting, could you provide a sentence or detail about how patients were involved, and what influence this had at each level of involvement please.

There is no mention of gaining ethical approval for the study, or the approval number in the methods section, I'm sure its an oversight, could you add it please.

P6 Data analysis

Whilst the whole paper is very well written and coherent, there is not enough detail or transparency about how the stages of analysis were completed, and the results were arrived at, this needs to be more explicitly included.

Results P7

P8 Table 1 What type of deprivation is being measured here, could you add more detail to the axes please

There is some unnecessary capitalisation mid-sentence throughout this section.

Its only a small study, could the authors add in brackets where they say many participants, how many, to give an idea of context.

This whole sub-section is very well written, with clear explanation, but at the moment it is not clear where these themes came from, this needs more explanation and linkage in the Methods section, good illustrative quotes.

Discussion P20

All the comments are accurate and relate to the paper, but it feels more like a summary rather than key points arising, and their implications,

what are the competing tensions? and how these concerns might be addressed - in several areas, e.g. societally, patient information, pregnancy related services, knowledge of health professionals… policies and publicity…where are the blockages, what needs to happen for services to communicate better… I'm not saying you need to resolve everything, but going beyond, to give specific ideas, pointers for better, more joined up practice, information for women ect, would lead to a stronger discussion section

what does it add, what needs to happen, what are the implications?

Conclusion P23

A really clear paper, with quality data, it would benefit from a stronger, bolder conclusion,

Reviewer #3: I read with great interest the Manuscript titled “Women's experiences of anal incontinence following vaginal birth: a qualitative study of missed opportunities in routine care contacts” (PONE-D-23-06127), which falls within the aim of this Journal.

In my honest opinion, the topic is interesting enough to attract the readers’ attention. Methodology is accurate and conclusions are supported by the data analysis. Nevertheless, authors should clarify some point and improve the discussion citing relevant and novel key articles about the topic.

Authors should consider the following recommendations:

- Inclusion/exclusion criteria should be better clarified.

- The Authors did not mention the sample size calculation for their study. It is essential to specify this data in order to guarantee an adequate significance of the results obtained by the Authors.

- The authors have not adequately highlighted the strengths and limitations of their study.

- What are the actual clinical implications of this study? it is important to report the results obtained by the authors in the context of clinical practice and to adequately highlight what contribution this study adds to the literature already existing on the topic and to future study perspectives

- Does this manuscript conform the Enhancing the QUAlity and Transparency Of health Research (EQUATOR) network guidelines? It would be mandatory to declare about this element.

- Authors should add further details to discuss the role of the perineum protection techniques during the management of the second phase of labour and the effect on the postpartum period (authors may refer to: PMID: 25909491; PMID: 24942141).

6. PLOS authors have the option to publish the peer review history of their article (what does this mean?). If published, this will include your full peer review and any attached files.

Reviewer #1: No

Reviewer #2: No

Reviewer #3: **Yes: **VERONICA LUMIA

---

## [Author Response · Author response to Decision Letter 0]

17 May 2023

Dear Editor,

Thank you for the opportunity to address reviewer comments on the manuscript titled ‘Women's experiences of anal incontinence following vaginal birth: a qualitative study of missed opportunities in routine care contacts’ (PONE-D-23-06127). 

We have addressed each comment in turn, and describe below our response to each comment. 

Comment from reviewer/ editor Response

We have been through the guidelines for the manuscript and made changes to formatting accordingly.

In the ethics statement in the Methods, you have specified that verbal consent was obtained. Please provide additional details regarding how this consent was documented and witnessed, and state whether this was approved by the IRB 

Consent was not documented, but interviews only proceeded if consent was obtained. Participants were given the option of providing verbal consent; audio recorded at the start of the interview and saved in a separate file on an encrypted university drive or written consent; returned to the research team by email and saved in a separate file to the interview file on an encrypted university drive.

All consent files were saved separately to the interview files and labelled with a unique participant ID to ensure anonymity. This process was in line with ethical approval gained for the study.

Thank you for stating the following in the Competing Interests section:

“I have read the Journal's policy and the authors of this manuscript have the following competing interests:

Professor Michael Keighley is Trustee and President of the MASIC Foundation but this has not impacted on conduct of study or publication, CEO of Keighleycolo Ltd involved with medical reports for court in which OASI patients have been seen but no conflict as this simply informed me of the consequences of the injury during one to one face to face consultations. Trustee Friends of Vellore UK also has helped to understand impact of OASI in the Asian community in India. Joint holder of two i4i NIHR grants for a device for the treatment of fistula in fact of value to some OASI patients.”

Thank you for this comment. 

In relation to the disclosed conflict of interest we confirm that this does not alter our adherence to PLOS ONE policies on sharing data and materials.

Professor Keighley has acted as an expert witness during his career, but at no point in the interviews were women asked about legal cases. Furthermore, Professor Keighley was not involved in interviewing women or recruiting women. There is no way of knowing if he acted as an expert witness for recruited women, but it would have had no impact on recruitment or data collection.

In your Data Availability statement, you have not specified where the minimal data set underlying the results described in your manuscript can be found. PLOS defines a study's minimal data set as the underlying data used to reach the conclusions drawn in the manuscript and any additional data required to replicate the reported study findings in their entirety. All PLOS journals require that the minimal data set be made fully available. For more information about our data policy, please see http://journals.plos.org/plosone/s/data-availability.

As this study used qualitative methods of data collection (interviews), there is no single data set, and replicability of analysis is not an appropriate consideration. 

Guidelines for the data policy suggest that excerpts of the transcripts relevant to the study are made available in an appropriate data repository. We do not feel this is something that we are able to do for this study, based on the ethical approvals and consent obtained for this study. We have ethical approval and consent from participants to include quotes, but any more than that is not covered by ethical approval or consent. We also believe it would be inappropriate to include more than short quotes, as it increases the chances that data is identifiable and therefore does not fulfil anonymity requirements. 

We therefore do not feel any additional data should be shared in a repository, but would be happy if readers want to contact the lead author to discuss any elements of the data. 

Your ethics statement should only appear in the Methods section of your manuscript. If your ethics statement is written in any section besides the Methods, please move it to the Methods section and delete it from any other section. Please ensure that your ethics statement is included in your manuscript, as the ethics statement entered into the online submission form will not be published alongside your manuscript. 

We have moved the ethics statement to the methods section of the manuscript in line with this comments. It has been deleted from its original position. 

We have checked over the reference list and are not aware of any retracted papers that have been cited in the manuscript. We have added in references where suggested by reviewers and these are highlighted by track changes.

Reviewer #1: Overall a well-structured, appropriately designed and original piece of research. Importantly, the value of this research in terms of the existing issue are set out clearly and potential avenues for improving experiences of women after childbirth injury are suggested without breaching the capabilities of qualitative research. An overview of the themes and subgroups in each would add clarity before the main text of the results. 

Thank you for the positive comments about the manuscript. 

We have added to the overview of the themes at the start of the results section, by adding in the subthemes for each theme.

Reviewer #2: Abstract P2

Clear, focused, check for unnecessary capitalisation in the results section 

We have checked the results section of the abstract, and removed any unnecessary capitalisation.

Intro P3

Line 54 ‘nulliparous women’, could the authors add a brief explanation of this term please, 

We have included a definition of what we mean by ‘Nulliparous women’ as ‘women who have never had a live birth.’ 

Line 55 ‘OASI injury’ could the authors add a brief explanation of this term please 

‘OASI injury’ is written in full in line 53, but we have added a brief description that is designed to compliment the explanation in the previous paragraph. 

Line 56 ‘Women of Indian and Pakistani origin are significantly more likely to sustain an OASI during childbirth (13,14, 58 15).’ Could you say briefly why this is please 

The reasons behind this risk factor are likely to be complex and multifactorial. We have added a sentence to explain this, and an appropriate reference. 

Line 61, could you add a sentence adding the implications please. 

We have added a line to explain that the increase in risk factors may have likely attributed to the rise in incidence

Lin 67, could you add a sentence to say why this is the case please, it’s a fascinating study, and not everyone will be a medical professional, or familiar with the field. 

We have added a short sentence to attempt to explain why urinary incontinence is discussed more than AI. 

Methods P4

P4 Line 73 date needed please 

We have included the date of the Ockenden review. 

P5 line 99 ‘Participants were also recruited via five hospitals across the UK’ could you say which countries, regions please 

We have added in the five regions of hospitals that participants were recruited from

P5 line . ‘Aiming for a maximum variation sample, women were recruited with varied backgrounds according to ethnicity, deprivation’ could you say why you wanted a variation sample, and what type of deprivation/how you were assessing deprivation please, 

We have explained why we were aiming for a variation within the sample, and have explained how we assessed deprivation. 

P5

Line 110 the PPI section is interesting, could you provide a sentence or detail about how patients were involved, and what influence this had at each level of involvement please. 

We have added to this section to show the different points PPI was used, and the benefits it had on the study.

There is no mention of gaining ethical approval for the study, or the approval number in the methods section, I'm sure its an oversight, could you add it please. 

Thank you for highlighting this. In line with the Editor’s comment we have moved the Ethical Statement into the Methods section rather than at the end of the manuscript. 

P6 Data analysis

Whilst the whole paper is very well written and coherent, there is not enough detail or transparency about how the stages of analysis were completed, and the results were arrived at, this needs to be more explicitly included. 

The data analysis followed the well-established method of Thematic Analysis, outlined by the work of Braun and Clarke. This follows six steps which have been detailed in the paper. We have added to this section to make it clearer and provided information for each step of the process.

Results P7

P8 Table 1 What type of deprivation is being measured here, could you add more detail to the axes please

We have added information as a note to the table to describe how deprivation was measured.

There is some unnecessary capitalisation mid-sentence throughout this section. 

We have checked the results section for unnecessary capitalisation and have corrected this. 

Its only a small study, could the authors add in brackets where they say many participants, how many, to give an idea of context. 

We have considered this point, but feel that as this is a qualitative study it is not usual to quantify findings in this way. The use of ‘many’ or similar was intended to imply that it was a strong feeling amongst the participants with similar responses. Further, we feel a sample size of 41 is a moderate size for a qualitative study.

This whole sub-section is very well written, with clear explanation, but at the moment it is not clear where these themes came from, this needs more explanation and linkage in the Methods section, good illustrative quotes. 

We have added some further explanation to how the themes were developed. We hope this will address this concern.

Discussion P20

All the comments are accurate and relate to the paper, but it feels more like a summary rather than key points arising, and their implications,

what are the competing tensions? and how these concerns might be addressed - in several areas, e.g. societally, patient information, pregnancy related services, knowledge of health professionals… policies and publicity…where are the blockages, what needs to happen for services to communicate better… I'm not saying you need to resolve everything, but going beyond, to give specific ideas, pointers for better, more joined up practice, information for women ect, would lead to a stronger discussion section

what does it add, what needs to happen, what are the implications? 

We have tightened up the implications for practice and research, and added some information about possible solutions to services. 

Conclusion P23

A really clear paper, with quality data, it would benefit from a stronger, bolder conclusion, 

We have reworked the conclusion section to tighten some of the messages. 

Reviewer #3: I read with great interest the Manuscript titled “Women's experiences of anal incontinence following vaginal birth: a qualitative study of missed opportunities in routine care contacts” (PONE-D-23-06127), which falls within the aim of this Journal.

In my honest opinion, the topic is interesting enough to attract the readers’ attention. Methodology is accurate and conclusions are supported by the data analysis. Nevertheless, authors should clarify some point and improve the discussion citing relevant and novel key articles about the topic. 

Thank you for the positive comments about the manuscript. 

We have revisited the discussion section (particularly the section about comparison with previous literature). We are confident that this section presents relevant and novel literature on the topic to compliment what is presented in the introduction section. We have also tightened up the strengths and limitations section which hopefully helps to clarify this section. 

Authors should consider the following recommendations:

- Inclusion/exclusion criteria should be better clarified. 

We have amended this section to show inclusion and exclusion criteria as a list to make it easier for readers. 

The Authors did not mention the sample size calculation for their study. It is essential to specify this data in order to guarantee an adequate significance of the results obtained by the Authors. 

As this is a qualitative study we did not conduct a sample size calculation. This is not part of qualitative methodology. In qualitative research, sample sizes are determined by data saturation; a sufficient number of interviews have been conducted whereby the research question is sufficiently answered, no further new information is evident, and no new themes are identified. 

The authors have not adequately highlighted the strengths and limitations of their study. 

We have reviewed the ‘Strengths and Weaknesses’ section of the manuscript and have further emphasised the strengths and limitations of the study. 

What are the actual clinical implications of this study? it is important to report the results obtained by the authors in the context of clinical practice and to adequately highlight what contribution this study adds to the literature already existing on the topic and to future study perspectives 

As with the earlier comment we have added to the implications section of the paper.

Does this manuscript conform the Enhancing the QUAlity and Transparency Of health Research (EQUATOR) network guidelines? It would be mandatory to declare about this element. 

In the methods we have noted that the Standards for Reporting Qualitative Research framework was used for reporting the research. The SRQR is listed on the EQUATOR network as one of the appropriate guidelines for qualitative research.

Authors should add further details to discuss the role of the perineum protection techniques during the management of the second phase of labour and the effect on the postpartum period (authors may refer to: PMID: 25909491; PMID: 24942141). 

We have added in a statement about perineal protection techniques in the introduction. We did not include it initially because the focus of this study is on the impact of the injury rather than the prevention of it. Thank you for the useful references. We have included one of them.

---

## [Decision Letter · Decision Letter 1]

13 Jun 2023

Women's experiences of anal incontinence following vaginal birth: a qualitative study of missed opportunities in routine care contacts

PONE-D-23-06127R1

Dear Dr. Parsons,

We’re pleased to inform you that your manuscript has been judged scientifically suitable for publication and will be formally accepted for publication once it meets all outstanding technical requirements.

Kind regards,

Antonio Simone Laganà, M.D., Ph.D.

Academic Editor

PLOS ONE

Additional Editor Comments (optional):

The authors performed the required corrections, which were positively evaluated by the reviewers. I am pleased to accept this paper for publication.

Reviewers' comments:

Reviewer's Responses to Questions

**Comments to the Author**

1. If the authors have adequately addressed your comments raised in a previous round of review and you feel that this manuscript is now acceptable for publication, you may indicate that here to bypass the “Comments to the Author” section, enter your conflict of interest statement in the “Confidential to Editor” section, and submit your "Accept" recommendation.

Reviewer #2: (No Response)

Reviewer #3: (No Response)

2. Is the manuscript technically sound, and do the data support the conclusions?

Reviewer #2: Yes

Reviewer #3: Yes

3. Has the statistical analysis been performed appropriately and rigorously? 

Reviewer #2: N/A

Reviewer #3: Yes

4. Have the authors made all data underlying the findings in their manuscript fully available?

Reviewer #2: Yes

Reviewer #3: Yes

5. Is the manuscript presented in an intelligible fashion and written in standard English?

Reviewer #2: Yes

Reviewer #3: Yes

6. Review Comments to the Author

Reviewer #2: The revisions make this a much stronger paper, its very well written and constructed, and an important topic which would benefit from dissemination and knowledge exchange activities.

There is just one aspect that needs a little more attention, then it's good to go. The explanation of the methodology is still too limited, it would benefit from more show, not just tell- how were the thematic analysis stages applied to this work, how did the themes become arrived at? Whilst it reads well really, its not clear how the authors went from inception to findings section.

There is still some unnecessary capitalisation in sentences, but this easily fixed.

Reviewer #3: I carefully evaluated the revised version of this manuscript.

Authors have performed the required changes, improving significantly the quality of the paper.

7. PLOS authors have the option to publish the peer review history of their article (what does this mean?). If published, this will include your full peer review and any attached files.

Reviewer #2: No

Reviewer #3: **Yes: **Veronica Lumia

---

## [Editor Report · Acceptance letter]

19 Jun 2023

PONE-D-23-06127R1 

Women's experiences of anal incontinence following vaginal birth: a qualitative study of missed opportunities in routine care contacts 

Dear Dr. Parsons:

I'm pleased to inform you that your manuscript has been deemed suitable for publication in PLOS ONE. Congratulations! Your manuscript is now with our production department. 

Kind regards, 

on behalf of

Dr. Antonio Simone Laganà 

Academic Editor

PLOS ONE